# Snow water equivalent can be monitored using RFID signal propagation

Mathieu Le Breton[1,2], Éric Larose[1], Laurent Baillet[1], Yves Lejeune[3], Alec van Herwijnen[4]

[1] Institut des Sciences de la Terre, Université Grenoble Alpes, CNRS, 38000 Grenoble, France
[2] Géolithe Innov, Géolithe, Crolles, 38920, France
[3] CEN-CNRM, Météo-France, CNRS, Saint-Martin-d'Hères, 38400, France
[4] WSL Institute for Snow and Avalanche Research SLF, Davos, 7260, Switzerland

*Correspondence to*: Mathieu Le Breton (mathieu.lebreton@geolithe.com)

**Abstract.** The amount of water contained in a snowpack, known as snow water equivalent (SWE), is used to anticipate the amount of snowmelt water that could supply hydroelectric power plants, water reservoirs, or sometimes cause flooding. This work introduces a wireless, non-destructive method for monitoring the SWE of a dry snowpack. The system is based on an array of low-cost passive radiofrequency identification (RFID) tags, placed under the snow and read at 865–868 MHz by a reader located above the snow. The SWE was deduced from the phase delay of the tag's backscattered response, which increases with the amount of snow traversed by the radiofrequency wave. Temperature was measured by the tag's internal sensor. Measurements taken in the laboratory, during snowfall events and over 4.5 months at the Col de Porte test field, were consistent with reference measurements of cosmic rays, precipitation, and snow pits. SWE accuracy was $\pm 18$ kg/m$^2$ throughout the season (averaged over 3 tags) and $\pm 3$ kg/m$^2$ during dry snowfall events (averaged over data from 2 antennas and 4 or 5 tags). The overall uncertainty compared to snow weighting was $\pm 10\%$ for snow density in the range $61-390$ kg/m$^3$. The main limitations observed were measurement bias caused by wet snow (which we discarded) and the need for phase unwrapping. The method has a number of advantages: it allows continuous measurement (1 min sampling rate in dry snow), it can provide complementary measurement of tag temperature, it does not require the reception of external data and it open the way towards spatialized measurements. The results presented also demonstrate that an RFID system can be used to remotely monitor the permittivity of a low-loss dielectric material with scientific-level accuracy, using propagation-based sensing.

# 1    Introduction

The snow water equivalent (SWE) of a snowpack represents the amount of water it contains (Fierz et al., 2009). SWE is used to anticipate the snowmelt water that will feed hydropower plants, fill water reservoirs, and potentially cause floods. It is also used to anticipate the risk of avalanches, to monitor the weight of snow on building, and for snow research. Many methods exist to monitor SWE but all have drawbacks (for review: Kinar and Pomeroy, 2015; Pirazzini et al., 2018; Royer et al., 2021). The methods based on sampling the snowpack (A. Denoth et al., 1984; Techel and Pielmeier, 2011) are destructive, require significant human resources and do not provide continuous measurements. Their automation, such as through the use of snow pillows (Beaumont, 1965), is technically complex. Snow models and satellite observations (Essery et al., 2013; Helbig et al., 2021; Tedesco et al., 2014) have a limited spatiotemporal resolution or suffer from limited accuracy. Radiation-based field methods (review, Royer et al., 2021) can conveniently and non-destructively monitor the SWE of a volume of snow. Among them, cosmic ray neutron probe (CRNP) (Kodama et al., 1979; Schattan et al., 2017) and gamma ray monitoring (GMON) (Choquette, Y. et al., 2013) are proven and mature methods, but they require specific instruments that are not only expensive but also complex to operate and calibrate (Royer et al., 2021). The dielectric permittivity of snow depends on its density and wetness, resulting in a direct relation between SWE and the delay of microwave transmission in the snow (Mätzler, 1987). Ground-penetrating radars can measure SWE from this delay (Bradford et al., 2009; Schmid et al., 2014, 2015), but they are expensive and their data is complex to process. GNSS (Koch et al., 2019, 2014) is a more convenient, light, compact, and low-cost method (Royer et al., 2021). Nevertheless, GNSS estimate the SWE with a daily sampling rate (Koch et al., 2019), needs GNSS satellite reception (Royer et al., 2021), and has a spatial resolution limited by the number of receivers.

Radiofrequency identification (RFID) technology also uses microwaves to identify goods equipped with passive tags. Passive RFID tags are produced by several billion units every year, allowing for low-cost tags (typ. €0.01–€20) and reading devices (typ. €2 k). A passive tag is basically an antenna and an ultra-low-power microchip. It is powered by a continuous wave (typ. around 865 MHz) emitted by the reader, which it modulates and backscatters to communicate to the reader. Recently, tags were developed with the capacity to sense their environment (reviewed by Costa et al., 2021), resulting in various applications in earth science (for review, see Le Breton et al., 2022). For example, tags were used to measure the temperature of the soil with an embedded sensor (Luvisi et al., 2016), and the presence of frost on the tag antenna through its change of impedance (Wagih and Shi, 2021). Tags can also be localized by measuring the variations of phase delay over time, between the reader and the tag (review by Xu et al., 2023). This technique was used to measure landslide displacements (Le Breton et al.,

2019; Charléty et al., 2022, 2023). Finally, Le Breton (2019) measured variations in the phase when the RFID signal transmits through snow and related this variation to snow density and thickness.

Therefore, we expect that an array of passive RFID tags placed under the snow may monitor SWE, using phase delay measurements. It may have a higher spatiotemporal resolution and lower cost than existing methods. We tested this hypothesis in the laboratory, during short snowfall events and throughout an entire season outdoors.

## 2 Method and instruments

### 2.1 Theory: from phase delay to SWE

The velocity of electromagnetic wave propagation in snow depends on the real part of its relative permittivity (Tedesco, 2015) that we call simply "permittivity". At the second order, the permittivity $\varepsilon'_s$ of dry snow at 10–1000 MHz depends on its density $\rho$ (in kg/m³) as follows:

$$\varepsilon'_s = 1 + a\rho + b\rho^2 \tag{1}$$

with the following approximate values for the empirical constants, a=1.7x10⁻³ m³·kg⁻¹, and b=0.7 x10⁻⁶ m⁶·kg⁻² (Tiuri et al., 1984). Each snow layer is considered, linear, isotropic, homogeneous, nonmagnetic ($\mu=\mu_0$), with a negligible scattering at 865 MHz. The dry snow has a very small conductivity (Mellor, 1977) and can be considered as a low-loss dielectric medium (Bradford et al., 2009). The wave velocity $v$ can then be expressed as a function of the snow permittivity $\varepsilon'$ and the velocity in a vacuum $c$ ($\approx 2.998 \cdot 10^8$ m/s) (Balanis, 2012):

$$v = \frac{c}{\sqrt{\varepsilon'}} \tag{2}$$

Roughly speaking, dry snow with density within 100−600 kg/m³ would have a permittivity within 1.1–2.3 (i.e., a relative velocity of 0.65–0.95). With the ray approximation, the phase $\phi$ (in radians) of a wave of frequency $f$ (in Hz), propagating two ways through a medium over a distance $d$ (in meters) equals:

$$\phi = \frac{4\pi f}{v} d \tag{3}$$

We represent the phase with the same sign as the time delay, for simplicity. Combining (1), (2) and (3), the phase variation when a homogeneous layer of dry snow replaces a layer of air can be approximated as:

$$\delta\phi = \phi_{snow} - \phi_{air} = \frac{4\pi f}{c}\left(1 - \sqrt{1 + a\rho + b\rho^2}\right)d \tag{4}$$

A first-order Taylor expansion on the density gives:

$$\delta\phi = \frac{2\pi f}{c}a\rho d \tag{5}$$

The expansion brings an error $< 0.5\%$ for 0–500 kg/m$^3$ density, which is negligible compared to SWE measurement uncertainty in general. Knowing that $SWE = \rho z$, with $z$ the snow depth, the variation $\Delta SWE$ due to the presence of multiple layers of snow, relates to the cumulative phase variation $\Delta\Phi$:

$$\Delta SWE = \frac{c}{2\pi fa}\Delta\phi \tag{6}$$

A phase shift of $\pi$ represents a SWE of 102 kg/m$^2$. In practice, the RFID reader measures the phase $\phi_{meas}(t) = \phi(t) + \phi_0(t) - k\pi$, with an offset $\phi_0$ and an unknown integer **k** causing a $k\pi$ ambiguity—$2k\pi$ with most recent
readers (Miesen et al., 2013). Appropriate instrumentation and processing workflow, presented in Sect. 2.2 and 2.3, reduce the unwanted variations of $\phi_0(t)$ and solve the phase ambiguity.

## 2.2   Instrumentation in the laboratory and outdoors

The experimental setup was designed to measure the increase in phase delay caused by the layers of dry snow formed between a reader antenna above the snow, and a tag below the snow. The SR420 reader (Impinj) emits and
receives a radiofrequency signal at selected frequencies (865.7, 866.3, 866.9 and 867.5 MHz), through an antenna. A slot antenna was used in the laboratory (Model IPJ-A0311-EU1, 5 dBi gain, linear polarization, 50°/100° Beamwidth at –3 dB), and two patch antennas were used outdoors (Model Kathrein 52020251, 12.5 dBi gain, linear polarization, 42°/42° Beamwidth at –3 dB, IP65). The tags (Survivor B, from Confidex, 2014) measure 155 × 26 × 14.5 mm and weigh 32 g each (see appendix 3). These tags are essentially passive, but the models used in this
study were assisted by a tiny battery (with several years' lifetime) to increase sensitivity and read-range. These devices are termed "battery-assisted" or "semi-passive" tags. The method is suitable for use with any passive backscattering tag (either battery-powered or batteryless), but not with active tags for which the phase is not synchronized between the receiver and the emitter. Each tag includes an antenna which converts the RF wave into a current, waking-up the microcircuit contained in the tag. The microcircuit (EM4325, from EM Microelectronic-

Marin) has ultra-low power requirements (<10 μW when interrogated), and embeds an integrated temperature

sensor with ±2.0 °C initial accuracy over −40°C to 60°C (Confidex, 2014), and ±0.25°C resolution and accuracy

over −7 to 0°C after calibration (see appendix 2). The material was chosen to reduce thermal influence on the phase

(Le Breton et al., 2017). During acquisition, the reader interrogates each tag sequentially for 30 ms, following a

standard RFID protocol (EPC-Gen2, 2015). When requested by the reader, a tag communicates its unique identifier

and any other data from its memory by backscattering and modulating the signal amplitude. For each tag, the reader

measures the Phase Difference of Arrival between the two modulated states of the incoming signal compared to

the continuous wave emitted (Nikitin et al., 2010). Here, this is termed the "phase". The modulated tag reflection

therefore distinguishes the static reflection from the environment and any signal from the tags that are not being

interrogated. Phase measurement is possible with backscattering communication because, unlike with classical

wireless communications, the reader can easily synchronize the emitted and received waves.

In the laboratory experiment, one reader antenna and one tag were placed 1 m above and 0.05 m below a 0.4 ×

0.4 m polystyrene box, respectively (Fig. 1). Step by step, layers of dry snow were added to the box, to form an

increasingly thick snow block, from no snow to approximately 0.24 m deep snow. The whole experiment was

performed in a cold room (-5 °C). The snow, collected outdoors was kept dry. It was sieved to add each new layer

to the box. After adding each layer, the snow surface was smoothed before measuring its thickness and the weight

of the entire snow block to estimate its density. The experiment was repeated with a snow density of 230, 275 and

330 kg/m$^3$, and a maximum snow depth of 0.24, 0.237 and 0.245 m, respectively. The snow density was increased

by repeatedly sieving the same snow but changing the mesh size.

The continuous field monitoring was installed during winter 2019–2020 at Col de Porte, France (alt. 1325 m). Col

de Porte is the French reference site for snow measurements and instrument testing (Lejeune et al., 2019), and is

operated by Météo-France's center for snow study (CEN). The numerous instruments present and manual surveys

conducted on this site provided an exhaustive dataset describing the snowpack and its environment throughout the

experiment (Fig. 2). Precipitation was measured by automatic weighing gauge, and used to estimate the variation

in SWE caused by snowfall events. The snow height was measured by a number of methods: automatic laser

instruments, manual surveys in snow pits, and manual inspection of a pole near the RFID tags. The SWE was

estimated automatically every day with a CRNP. The air temperature was measured by a meteorologic station, and

the snow surface temperature was monitored by infrared sensors. A webcam collected images of the measurement

sites every hour, which were used to monitor the melt surrounding individual tags.

In the field installation, two vertical arrays of tags—comprising 12 and 11 tags each—were planted on the ground.
The tags were placed 4−169 cm and 8−158 cm above ground, respectively, with 0.15-cm spacing (see Fig. 3). The tags were supported by a 0.05-m-diameter and 1.70-m-high PVC tube, equipped with multiple 0.15-m-long and 0.02-m-thick horizontal plastic arms. The arms were supported from below and the PVC tubes were maintained by rigging strings, to avoid movement. The two reader antennas were placed above the tags, 4 m from the ground. The reader antennas were supported by a metallic arm attached to a large vertical metallic pole, 3 m from the tag support.
The acquisition lasted from 2019-10-22 to 2020-03-27. Experiments initially focused on four snowfall events, during which the top layers of snow remained entirely dry, then the SWE was computed over the whole winter, using the workflow described below.

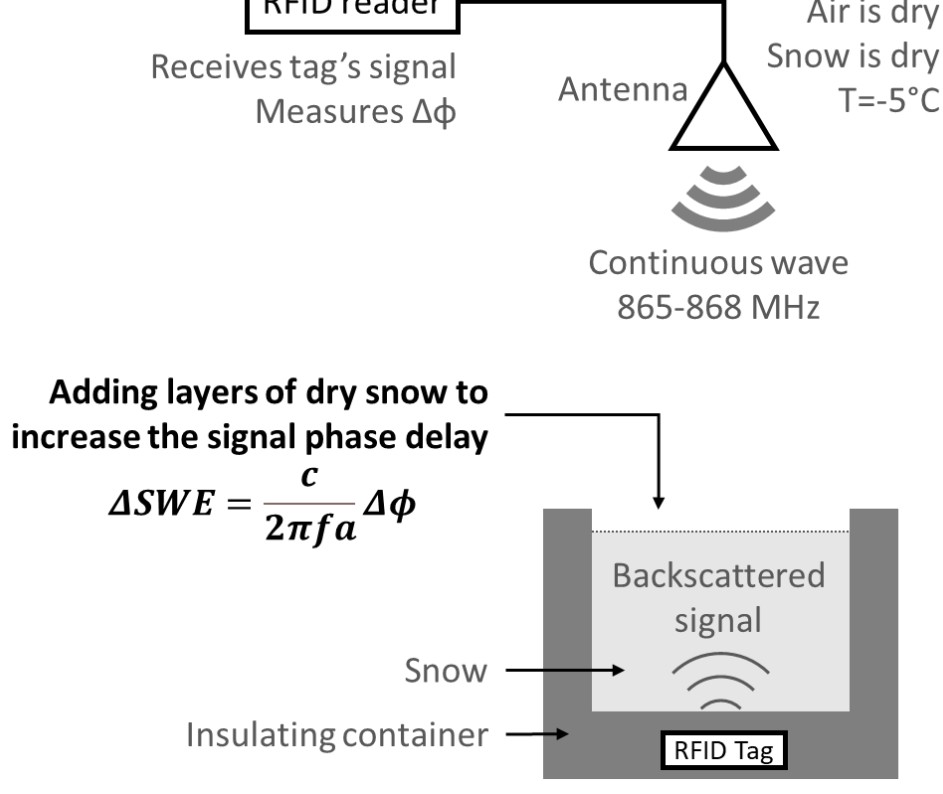

Fig. 1: Laboratory setup to simulate new layers of snow, and validate the SWE estimation from the change of phase delay between
140 the tag and the reader antenna.

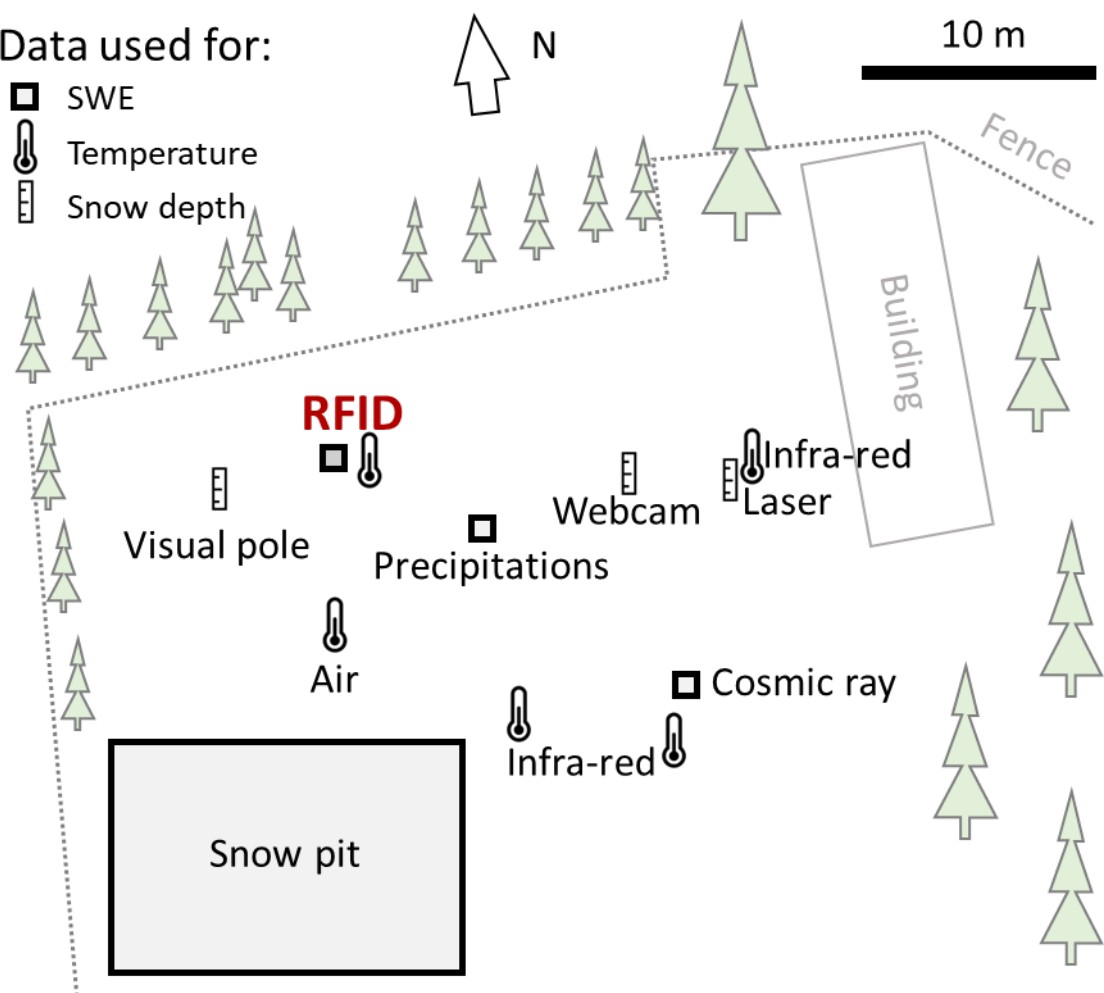

**Fig. 2: Site of Col de Porte, highlighting the positions of the reference instruments. Modified from Lejeune et al. (2019)**

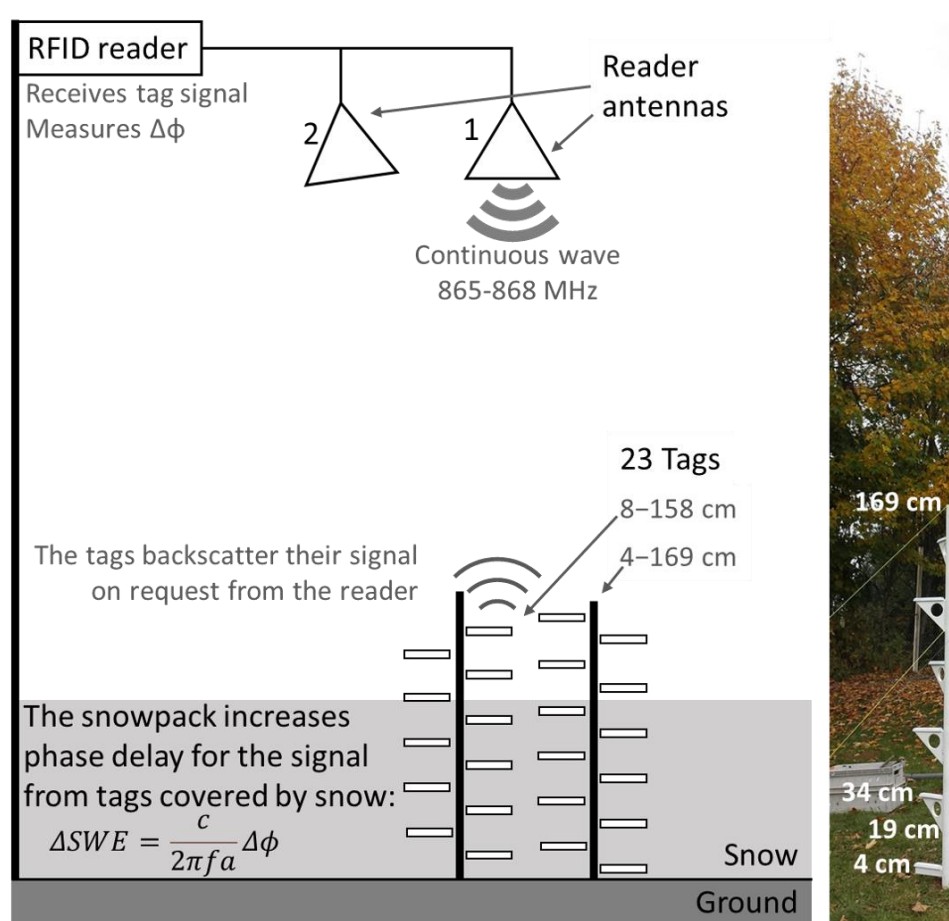
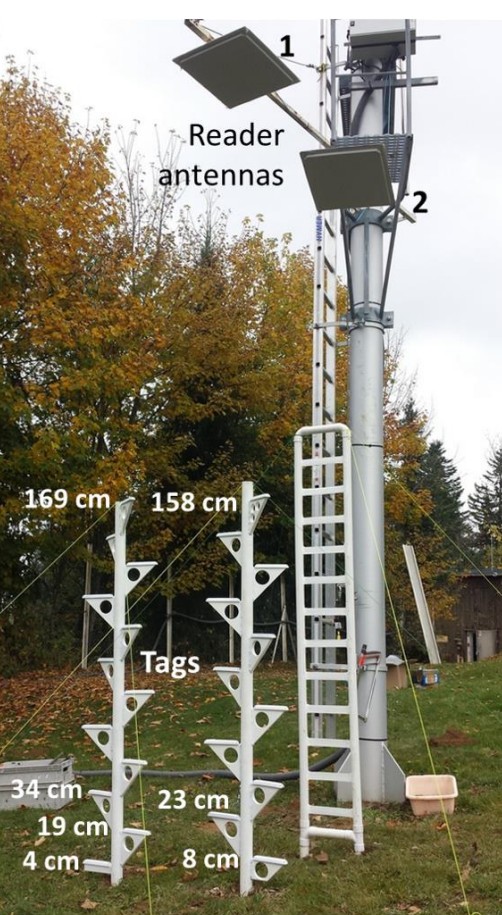

**Fig. 3: Outdoor experimental RFID setup at Col de Porte.**

## 2.3 Workflow to compute SWE outdoors

The SWE was computed over the season using the following steps. The choices and adaptations specific to this study are marked in italics and further discussed in Sect. 4.

**1) Data selection.** Phase data were separated for each combination of tag, reader antennas and frequencies available, to select the data to be processed. The tags covered by the snow are selected from their daily temperature variation that is smaller than with tags in the air (Reusser and Zehe, 2011) (see temperature data on Fig. A3).

*We selected for individual events of dry snowfall based on dry snow criteria (step 3), on (1) 2019-12-11, (2)2019-12-12/13, (3) 2020-01-10 and (4) 2020-02-27. We used only the tags covered by snow, at heights of 4−23 cm for events 1 and 3, and at heights of 4−34 cm for events 2 and 4 (Sect. 3.2).*

*We split the season in three periods, starting on (1) 2019-10-23 (2) 2019-12-19 (3) 2020-02-03. We used tags at height of 4 cm for the period 1, and at heights of 4−19 cm for the periods 2−3 (Sect.3.3).*

**2) Phase unwrapping**. The phase was unwrapped to cumulate phase variations over time to solve its $k\pi$ ambiguity (equivalent to $k\times102$ kg/m$^2$ of SWE for dry snow), with the hypothesis of data continuity.

*We combined the phases of the four frequencies available. We also removed the fast variations of phase using a complex domain averaging over 3 minutes, unwrapped the smoothed phase, then reintroduced these variations (see Charléty et al., 2023).*

**3) Dry snow selection**. The periods of dry snow were selected to ensure that the snow permittivity was influenced only by its density (needed for eq. 6) and not by its liquid water content (Tiuri et al., 1984).

*For most of the season, we identified and removed wet snow periods from their phase delay which displayed rapid and non-mononotous fluctuations over the day, typically from 08:00 to 24:00. It was also validated from, the temperature of the snow surface < 0°C measured by infrared and by tags close to the surface, and from air temperature <0°C when precipitation occurred. After 2020-03-03, the snowpack rarely refroze completely during the night, so we picked only the period of driest snowpack (with a local phase maximum), typically 06:30–07:00. We also identified the four individual events of dry snowfall.*

**4) SWE conversion**. The variation of phase was converted into a variation of dry snow SWE using eq. (6)

**5) Recalibration in case of technical issues.** Sometimes, recalibration may be required to compensate for a technical issue (Charléty et al., 2023). *The alteration of the snowpack just above the tags can cause a local SWE offset and would need to be compensated.* In addition, after a long data gap due to technical

issues, the phase ambiguity might need to be resolved. In this case, the variation of $k$ occurring during the gap could be estimated with an independant method which accuracy is below half the ambiguity.

However, we recalibrated twice the SWE to compensate for accelerated melting around the tag supports during warm periods with rainfall (see appendix 5). This recalibration resulted in three distinct periods in Fig. 6, with two periods recalibrated based on snow pit measurements (marked as "ref"). We encountered no data gap causing ambiguity issues here.

**6) Spatial averaging.** The error caused by multipathing interferences can be reduced by computing the mean data between the different tags and antennas.

We used the tags selected in step 1, measured from two antennas during the snowfall events, and from one antenna, with the highest signal strength, during the season.

**7) Time averaging.** Data were averaged at the desired sampling duration.

We kept the 1 min time sampling for the snowfall events (Fig. 5). We averaged over 12 h for the entire season to account for the discarded periods of wet snow (Fig. 6).

The tag temperature sensors were also calibrated at 0°C when surrounded by wet snow (see appendix 2).

# 3    Results of SWE measurements

## 3.1    Laboratory experiments

Laboratory results confirmed that the variation in SWE estimated from the RFID phase (Fig. 4, solid line) was
consistent with the SWE estimated from snow weights, over the complete cumulated layers (Fig. 4, dashed lines).
This result was verified for snow density of 230, 275 and 335 kg/m$^3$, corresponding to snow permittivity of 1.43,
1.51 and 1.64, respectively (eq. 1). The estimated SWE oscillated depending on the snow depth, within $\pm10$ kg/m$^2$
of the value obtained by weighing the snow. The spatial period corresponded to half a wavelength in the snowpack
(0.135–0.145 m for the highest–lowest density, respectively), which strongly suggests that it results from fringes
of multipath interference caused by reflection of waves at the air-snow interface (Le Breton, 2019). In conclusion,
the method worked well under controlled conditions, with $\pm10$ kg/m$^2$ accuracy for a single tag-antenna
combination, and an error that could mostly be attributed to multipathing.

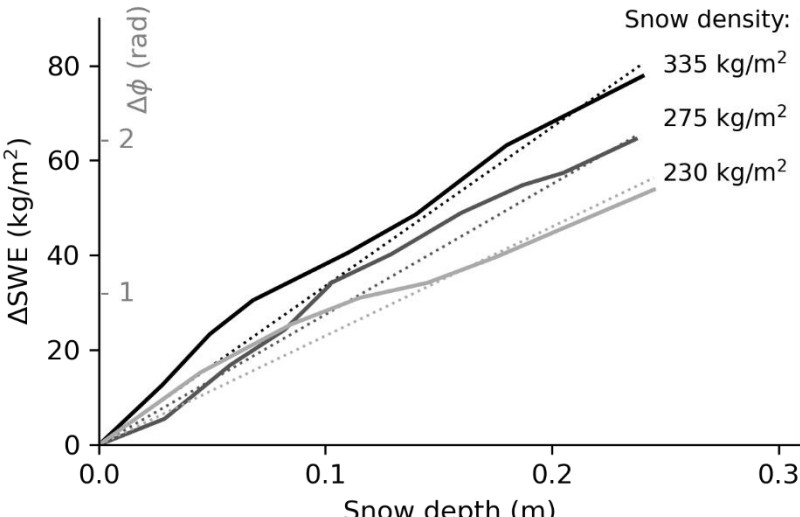

**Fig. 4: Cumulated variations of SWE obtained from RFID phase measurement (solid lines) and weighing (dashed line), as a function
of the thickness of the snow block, for three densities.**

## 3.2 Snowfall events

For each dry snowfall event selected, the depth of snow and the cumulated precipitation—which equals the SWE
when no melting occurs—were compared to the RFID measurements made every minute (Fig. 5). The SWE
estimated from a single tag-antenna combination exhibited dispersion up to $\pm30$ kg/m². The dispersion was different
for each event, each tag and each antenna, suggesting that the method is sensitive to tag position, antenna position
and the snowpack's geometry. For example, on 2019-12-11, the 18-cm and 23-cm-high tags provided biased SWE
only from antenna 1. The dispersion is consistent with the expected influence of multipathing (see discussion, and
220 appendix 6). The average SWE estimated from all the tags and antennas (Fig. 5, black line) was very close to the
cumulated precipitation (black squares), with a full-amplitude error up to $\pm3$ kg/m² (details on appendix 1). In
conclusion, the RFID array prove efficient to measure SWE accurately with 1 min resolution during short periods.

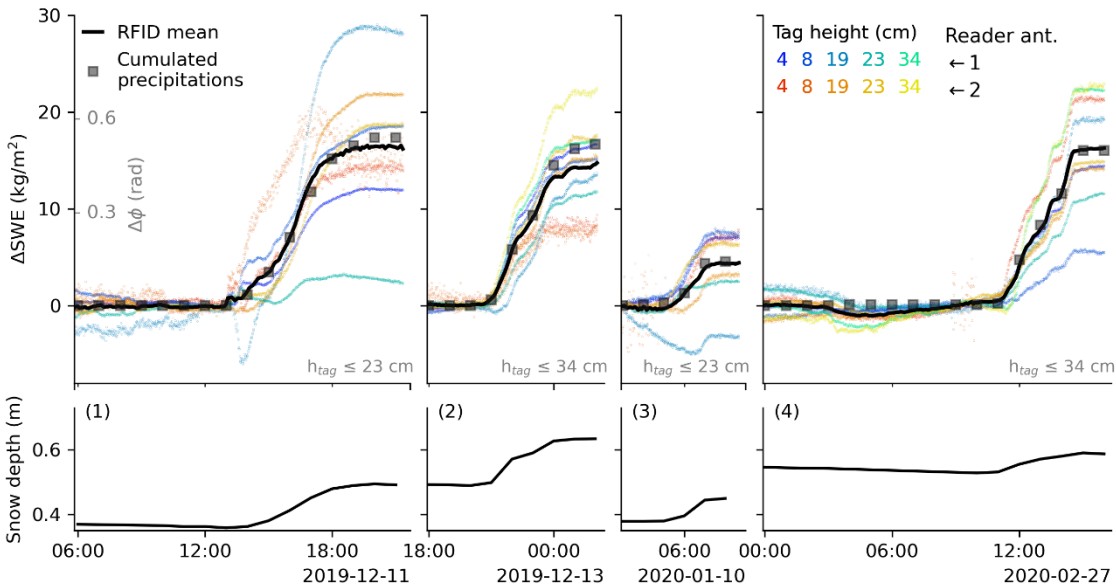

**Fig. 5: Increase of SWE measured over the course of four dry snowfall events, using single tags and antennas from the RFID array**
**(see Fig. 3) (in color), the median value of the array (black line) and precipitation measured by weighting gauge (gray squares). The
snow depth measured by laser is also displayed.**

## 3.3 Entire season

Over the entire season, the SWE estimated by RFID (Fig. 6, in red) is consistent with the CRNP and snow pit
measurements (in gray and black). During snowmelt periods, around 2019-11-27 and after 2020-03-08, RFID
sensing appeared to be more accurate than CRNP, which is influenced by water present in the soil (Sigouin and Si,
2016). Given the accuracy of CRNP (which has its own limitations) and the spatial heterogeneity in the snowpack,
we considered the results close enough to validate the RFID method. We measured an uncertainty of ±18 kg/m²
compared to snow pit (see appendix 1).

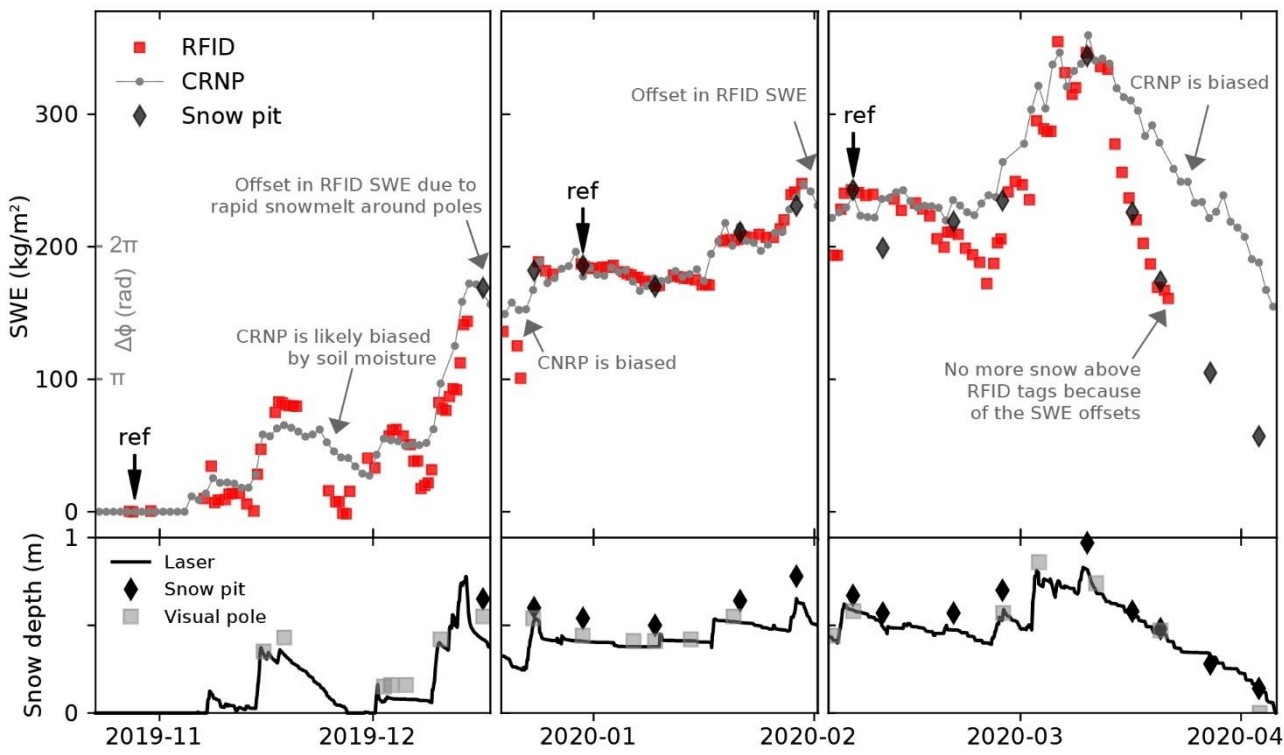

Fig. 6: SWE measurements for the three periods with RFID (top), keeping only the driest snowpack time windows. CRNP and snow
pit survey (bottom.) Snow depth was measured at three locations using a laser sensor, manual surveying and a visual pole. During
the first period, the data were derived only from the 4 cm high tag, due to the shallow snow depth. In subsequent periods, the data
from the three lowest tags (4 cm, 8 cm and 19 cm) were averaged. For each period, the SWE RFID estimation was calibrated relative
to a reference SWE based on a manual measurement, indicated by the "ref" arrow.

### 3.4    SWE measurement accuracy compared to weighting

The difference between the SWE measured by RFID and by weighing was $\pm 10\,\text{kg/m}^2$ in the laboratory, $\pm 3\,\text{kg/m}^2$ during short snowfalls, and $\pm 18\,\text{kg/m}^2$ during the last two periods of the season (details on Fig. A1). We did not compare the measurements with CRNP values, as we considered it not to be accurate enough to represent ground truth data. Laboratory measurements were not the most accurate, because the single combination of tag and antenna made them more sensitive to multipathing. On the contrary, the most accurate measurements occurred during snowfall, with an averaging over 4−5 tags and 2 antennas. Therefore, increasing the number of tags and antennas is the most important factor when seeking to increase accuracy, with most inaccuracies caused by multipathing.

The snow density (Fig. 7), computed as the SWE normalized relative to the snow depth, indicates that the RFID measurements occurred on 61−390 $\text{kg/m}^3$ snow density. The role of settling (Helfricht et al., 2018) was partially compensated in the density calculation,  by removing the trend of snow depth decrease (visible on events 1 and 4) obtained after precipitation. Both RFID and weighting SWE used the same snow depth, so the relative error is unchanged. Overall, RFID measurements fitted within a 10% relative uncertainty compared to weighting, for 61 $\text{kg/m}^3$ to 390 $\text{kg/m}^3$ density.

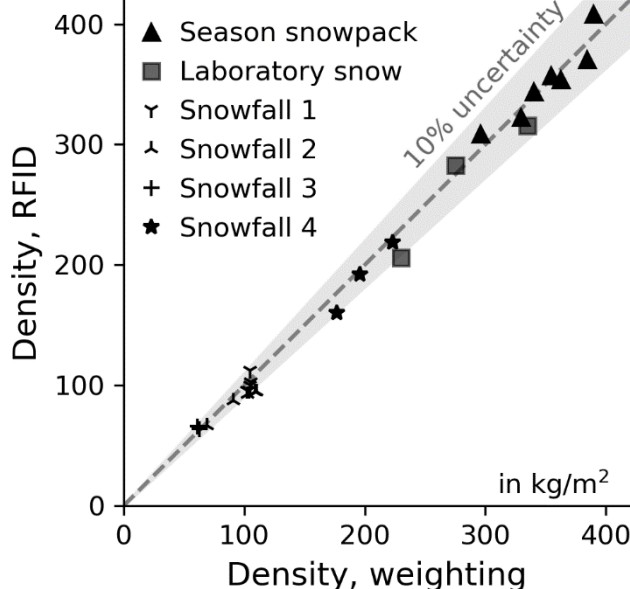

**Fig. 7: Comparison of snow density estimated from the SWE obtained by weighting or by RFID SWE (for a known thickness). The RFID method works with fresh and compacted snow, from 65 kg/m³ to 390 kg/m³ density.**

## 4 Discussion

**Table 1: Evaluating RFID method using criterias of Royer (2021)**

| Criteria | RFID SWE performances in this study |
|---|---|
| Uncertainty | $\pm10\%$ and $\pm18$ kg/m$^2$ compared with weighing |
| SWE$_{max}$ | 3000 kg/m$^2$ (theoretical value, for 6% volume liquid water content) |
| Other measured data | T°C vertical gradient— may also measure liquid water content in the future |
| Depends on external data | No. No need for satellite reception, ancillary data, or data from an external station |
| Typical sampling rate | Continuous — except for wet snow that currently bias measurements |
| Area of snow measured | <1 m$^2$ |
| Price | Should be similar to GNSS |
| Power consumption | 7 W with 1 min sampling — may be optimized |
| Advantages | Mass-market availability of the hardware (vs. CRNP&GMON) |
| | Tag array improves accuracy and enables spatialization (vs. all) |
| | Works both with deep and shallow snowpack (vs. CRNP&GMON) |
| Limitations | Requires continuous measurements for phase unwrapping. |
| | Biased by wet snow —may be corrected in the future |
| Maturity | RFID hardware and software in the field are reliable. |
| | Developments needed on the tag array and on data processing automatization |

We compared the RFID performance to other non-destructive SWE monitoring methods described as mature by Royer et al. (2021): CRNP, GMON and GNSS. We omitted multifrequency radar because its signal does not transmit in wet snow due to a severe attenuation at 24 GHz. *The uncertainty* of $\pm10\%$ and $\pm18$ kg/m$^2$ between RFID and weighing was similar to that obtained with the other methods, between 9% and 15% (Royer et al., 2021). However, estimating the uncertainty is difficult because the snowpack is heterogeneous, and because no data

represents the ground truth, rigorously speaking (Royer et al., 2021). *The sampling rate* used was <1 s in the raw data (the reader interrogates a tag every 30 ms), 1 min during snowfall events to reduce random noise, and 12 h during the full season due to the discarded wet snow period (wet snow could be corrected in the future, as discussed later). The 1 min sampling rate is considerably better than the typical 1-day rate possible with CRNP, GMON and GNSS. *The maximum SWE* measurable might be around 3000 kg/m$^2$, based on our theoretical estimation (discussed below). *The complementary measurements* include vertical temperature gradient measured by the tags. It might also include the liquid water content in the future, based on signal attenuation measurements (Koch et al., 2014) (discussed below). *The RFID method is not dependent on external data*, it thus outperforms the other methods which need either satellite reception (Koch et al., 2019), cosmic ray flux reference data, or atmospheric humidity and barometric pressure (Sigouin and Si, 2016). *The area covered* was <1 m$^2$, comparable with the GNSS method but much less than the GMON and CRNP methods, which sense the snowpack all around. Sensing the snowpack over a larger area is generally preferable to avoid localized snowpack variability (e.g., local snowmelt caused by the installation, and natural differences due to wind, topography, shade, etc.). Local sensing could be useful, however, if it was spatialized. *The price* of a fully operational system is currently unknown because it is not yet commercialized. We can only say that the reading station accounts for most of the cost, and that the cost of tags is negligible. We can reasonably anticipate a price within the range of existing methods, i.e., from €8 k to €17 k in 2021 for the sensor alone (Royer et al., 2021) (excluding installation, power, telecommunication, maintenance, etc.). *The method has three advantages*. First, the RFID hardware is a commodity, produced at industrial scale using interoperable standards, like GNSS, but in contrast to GMON and CRNP. This ensures a better balance between cost, reliability and long-term availability than likely with custom sensors. Second, the fact that an array of tags can easily be used increases the accuracy, and may enable spatialization. Third, the measurements are not biased by soil moisture, unlike GMON and CRNP, making the method more suitable for monitoring shallow snow depths when melt snow infiltrates the soil (using RFID measurements when snow is refreezing to reduce melt snow bias). *The method has two limitations today*. First, the phase must be unwrapped to deal with ambiguity. This requires an efficient, and potentially complex, unwrapping algorithm (Charléty et al., 2023), and continuous measurements to avoid large swathes of missing data during which the SWE could vary by more than ±102 kg/m$^2$. Second, measurements are biased by wet snow, which led us to discard this data. These limitations, discussed in the next paragraphs, might be mitigated in the future. *RFID hardware is mature*, and the acquisition system (for instance provided by Géolithe) has been continuously improved as part of its use to monitor several landslides since 2017 (Le Breton et al., 2019; Charléty et al., 2022, 2023). More developments could improve the tag array, fully

automatize data processing, reduce power consumption, and mitigate the method's limitations aforementioned. *In conclusion, the RFID method matches modern non-destructive snow sensing methods*, providing several advantages: no external data needed, high temporal resolution, temperature gradient data, large industry, not affected by soil moisture. Its limitations (it needs phase unwrapping and it is biased by snow wetness—could be mitigated in the future.

The issue of multipathing interference, for example, was mitigated in this study using tag arrays. Multipathing is a major challenge with RFID, because interferences from the waves reflected by the environment can reduce the received signal strength (Lazaro et al., 2009) and alter the phase (Arnitz et al., 2012). In addition, the snowpack strongly influences multipath patterns, as seen with GNSS reflectometry (Larson et al., 2009) and GPRs (Espin-Lopez and Pasian, 2021; Kulsoom et al., 2021). A few centimeters of snowpack can modify the phase and signal strength of fixed tags above the ground up to $\pm 1.5$ rad and $\pm 10$ dB (Le Breton, 2019) (See Fig. A7). A first potential mitigation approach is to remove or hide reflectors (e.g., Lucas et al., 2017). Removing the vertical tag array would reduce the number of reflectors, but the snow would still create strong interference. Another mitigation approach could be to model the entire environment (Hechenberger et al., 2022) to correct the phase, using propagation models in a snowpack (Proksch et al., 2015). However, this is highly complex and dependent on the environment model, and we found no mention of any such approach in RFID localization methods (Xu et al., 2023). Another mitigation approach would be to increase the bandwidth (Arnitz et al., 2012), but RFID bandwidth is narrow, within 1.8 MHz to 26 MHz for frequencies around 900 MHz, depending on regional regulations (e.g., ETSI-EN 302-208; FCC part 15). Finally, multipathing can be mitigated using an array of tags and reader antennas (e.g., Grebien et al., 2019). This is the option we used here. During snowfall events outdoors, we reduced the measurement bias from 30 kg/m$^2$ to 3 kg/m$^2$ by averaging measurements over 8 to 10 combinations of tags and antennas in different locations. Over the entire season, qualitatively, the SWE measured was more stable when averaged over 3 tags in periods 2&3, than over a single tag in period 1 (Fig. 6). In conclusion, using an array of tags and reader antennas efficiently mitigates RFID multipathing uncertainty.

The wet snow bias, in contrast, has yet to be mitigated. The increase of liquid water content in the snow can increase its permittivity (e.g., Bradford et al., 2009; Tiuri et al., 1984), increasing the phase delay and leading to overestimation of the SWE. For example, for a snow density of 500 kg/m$^3$, a liquid content increasing to 6% would increase the permittivity from 2 to 2.7, resulting in a +35% overestimation of the SWE. In addition, liquid water near the tag can increase the phase by changing the impedance of its antenna (Caccami et al., 2015; Le Breton et al., 2017). This effect would result in strong phase changes if ice melting occurs on the tag (Wagih and Shi,

2021). The combination of both effects explains the peaks of phases that occurred almost every day with sun light, or with wet precipitation (visible on Fig. A5). We manually discarded these data to retain the best possible SWE accuracy. Should we keep the discarding method in the future, the picking of wet periods could be automated based on a combination of signal loss (e.g., Koch et al., 2019), stable 0 °C temperature (e.g., Cheng et al., 2020; Dafflon et al., 2022; Reusser and Zehe, 2011), and phase peak recognition. Alternatively, the liquid water content present in the snowpack might be measured from the signal attenuation (e.g., Koch et al., 2014), to allow its influence on the phase to be corrected. In conclusion, the bias due to wet snow led us to discard the data from periods when the snow was wetter, and this limitation could be overcome in the future.

Phase ambiguity and unwrapping is another typical issue with RFID localization and sensing based on the phase. First, it requires an adequate unwrapping algorithm that is not influenced by short spurious noise in the phase (Charléty et al., 2023). In our experience, despite the use of advanced algorithms, some unwrapping issues can remain (phase jumps of $\pm\pi$). These are easily identified and corrected by human intervention—we made three corrections in our time series over the season. To overcome this need for manual intervention, one possible solution would be to exploit the tag array in the unwrapping algorithm. A second issue is that for unwrapping to proceed correctly, the phase must not vary by more than its ambiguity between two consecutive measurements (equivalent to $\Delta$SWE $\pm102$ kg/m$^2$ with modern readers). The method therefore requires continuous acquisition, without large data gaps. If some data is missing, the phase ambiguity would have to be solved using an independent method to estimate the unmeasured SWE variation with an uncertainty of less than $\pm102$ kg/m$^2$. Absolute localization methods based on tag arrays (Xu et al., 2023; Le Breton and Grunbaum, 2023) could also be investigated. In conclusion, the phase ambiguity is a limitation of the RFID method, because it requires a robust unwrapping algorithm and continuous data.

In contrast, measuring the snow temperature gradient using sensors in the tags (see data on appendix 2) is a definite advantage. We measured an accuracy of $\pm0.25$ °C within $-7$°C to 0 °C, after calibration, and saw no visible drift at 0°C for 3 months (see appendix 1). That is in line with the $3\sigma$ accuracy of $\pm0.2$°C to $\pm1$°C near 0°C, and of 0.5°C to 1.5°C within $-10$°C to 30°C, on hundreds of battery-assisted tags (Jedermann et al., 2009). It is also similar to the accuracy after calibration of $\pm0.2$°C near 37°C with commercial batteryless tags (Camera and Marrocco, 2021). In the snow, except for a few studies that reported a better accuracy or spatial resolution (e.g., Dafflon et al., 2022; Cheng et al., 2020), most studies used vertical temperature data that was measured with similar performances, to estimate other physical indicators of the snowpack. Therefore, our temperature data may also be

used to estimate the snow depth (Reusser and Zehe, 2011), water content (Marchenko et al., 2021), heat transfer (Brandt and Warren, 1997), thermal diffusivity (Oldroyd et al., 2013), and latent heat (Burns et al., 2014).

The SWE remained <350 kg/m$^2$ in this study. We can estimate the maximum SWE measurable using the basic theory of microwave propagation in snow (e.g., Koch et al., 2014; Le Breton, 2019; Steiner et al., 2019). Its value is limited by the tag's maximum read-range in the snowpack (see the influences on the read range on: Le Breton et al., 2022). This value depends mostly on the RFID hardware (Nikitin and Rao, 2006) and on the signal attenuation by the snow liquid water content (Koch et al., 2014). A snow with 500 kg/m$^3$ density and 6% of its volume containing liquid water would have a permittivity of $2.63+0.053\,j$ (Tiuri et al., 1984). The attenuation coefficient $\alpha = \frac{1}{2c}\frac{\varepsilon''}{\sqrt{\varepsilon'}}2\pi f$ (Bradford et al., 2009) (in m$^{-1}$), equivalent to $L_{dB} = -\frac{20}{\ln(10)}\alpha$ (in dB/m), leads to a reduction of signal strength $\Delta P_{dB} = L_{dB} \times 2\,h = $ **6.6 dB** $\times$ **h** in this snow. At normal incident angle, the loss due to reflection at the air-snow interface (around 0.5 dB) is much smaller than bulk attenuation. The other factors (multipathing, antenna coupling, reflectors within the snowpack) should be secondary compared to propagation attenuation if an appropriate tag array design is used . The maximum read-range in snow r$_{max,\,snow}$ is computed relatively to the maximum read-range in air r$_{max,\,air}$ using $\left(\frac{r_{max,air}}{r_{max,snow}}\right)^4 = 10^{\frac{\Delta P_{dB}}{10}}$ . The maximum SWE is the antenna height for which the power budget available in air equals the loss in the snowpack. These calculations result in a maximum theoretical SWE of 3000 kg/m$^2$ (6 m snow depth) for a battery-assisted tag readable at 60 m in the air (e.g., Survivor B), and 2250 kg/m$^2$ for a batteryless tag readable at 27 m in the air (e.g., Survivor M780). The real maximum SWE may be lower in practice, but nevertheless remains in the range of the GNSS limit of 2000 kg/m$^2$ (Royer et al., 2021).

Permittivity sensing had been demonstrated  with RFID tags, either by measuring the variations in tag antenna impedance (Bhattacharyya et al., 2010; Manzari and Marrocco, 2014; Caccami et al., 2015; Caccami and Marrocco, 2018) or by connecting a sensor to the tag (e.g., Fonseca et al., 2018). But these methods can characterize only the material in contact with the tag. Besides, their accuracy was lower than standard scientific instruments, due to the tag's limitations. In terms of accuracy, only the localization of tags in the air by the reader (see review: Xu et al., 2023) could match the accuracy of the standard techniques such as GNSS. Like localization, our sensing method is based on wave propagation, occurring, however, in another medium than air. We demonstrated that propagation-based sensing can measure the permittivity of material bulk, remotely, with scientific-level accuracy. In future, this method could also be applied to other materials, such as vegetation (Le Breton et al., 2023).

Finally, any tag can be used with this method. It needs only a reader that can read the phase of the received signal. If the read range—frequency-dependent in wet snow—is sufficient, the method should also work with harmonic tags (Mondal et al., 2019) already used under the snow (Mike Stanford, 1994; Grasegger et al., 2016), and with chipless tags (Barbot and Perret, 2018).

## 5    Conclusions

We introduced a method to sense the snow water equivalent of a snowpack, which works with standard radiofrequency identification devices. Its performance was similar to mature, non-destructive, scientific-level snow sensing methods (GNSS, gamma ray monitoring and cosmic ray neutron counting), with the accuracy of $\pm10\%$ or $\pm18$ kg/m$^2$ (see all criteria listed in Table 1).

In terms of advantages, the RFID method is fully independent and does not require external data or devices (e.g., GNSS reception, temperature and pressure sensors, incoming cosmic ray fluxes). It measures data continuously with a high temporal resolution <1 min in dry snow. Provided the usage of temperature-sensing tags, it can also measure the snow temperature gradient, with the accuracy of $\pm0.25$ °C at around 0 °C. It is not affected by soil moisture content. The long-term availability of the devices is supported by the large RFID industry.

The main limitation of the RFID method is its uncertainty when dealing with wet snow. This uncertainty led us to discard wetter snow periods, but it may be corrected in the future using independent liquid water content estimations. The need for continuous data to avoid phase ambiguity (equivalent to $\pm102$ kg/m$^2$ SWE) is also inconvenient. This difficulty can potentially be solved with advanced localization techniques, but further investigation would be needed.

In terms of RFID sensing, we showed that an array of tags can sense a material's bulk permittivity remotely using propagation-based sensing. The results presented demonstrate that RFID propagation-based sensing systems can achieve the accuracy of scientific-level instruments.

Future developments should aim to improve tag array design, correct the bias caused by wet snow, investigate phase solving methods, and automate data processing.

## 6  Acknowledgments

We acknowledge funding from the French National Agency for research (ANR) through the LABCOM Geo3iLab project (ANR- 17- LCV2-0007-01), and support from Géolithe Innov in running the experiments. We thank G. Scheiblin from ISTerre, and J. Roulle and the technical staff from Centre d'Étude de la Neige, for support on experiments. We thank EDF (Électricité de France) for providing access to their CRNP/NRC data. We thank M. Gallagher for the proofreading. We thank M. Dumont, F. Karbou, D. Jongmans, E. Rey, F. Guyoton, A. Belleville and P. Carrier, for their interest in our project and for fruitful discussions in the early stages and throughout the process.

## 7  Contributions

MLB developed the theory, analyzed the data, and wrote the manuscript. MLB, LB, EL conceptualized the method and acquired funding. EL, MLB, AVH and YL designed, planned and performed the experiments. All the authors discussed the results and contributed to the manuscript.

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

## Appendix 1: Uncertainty between SWE measured by RFID and weighting

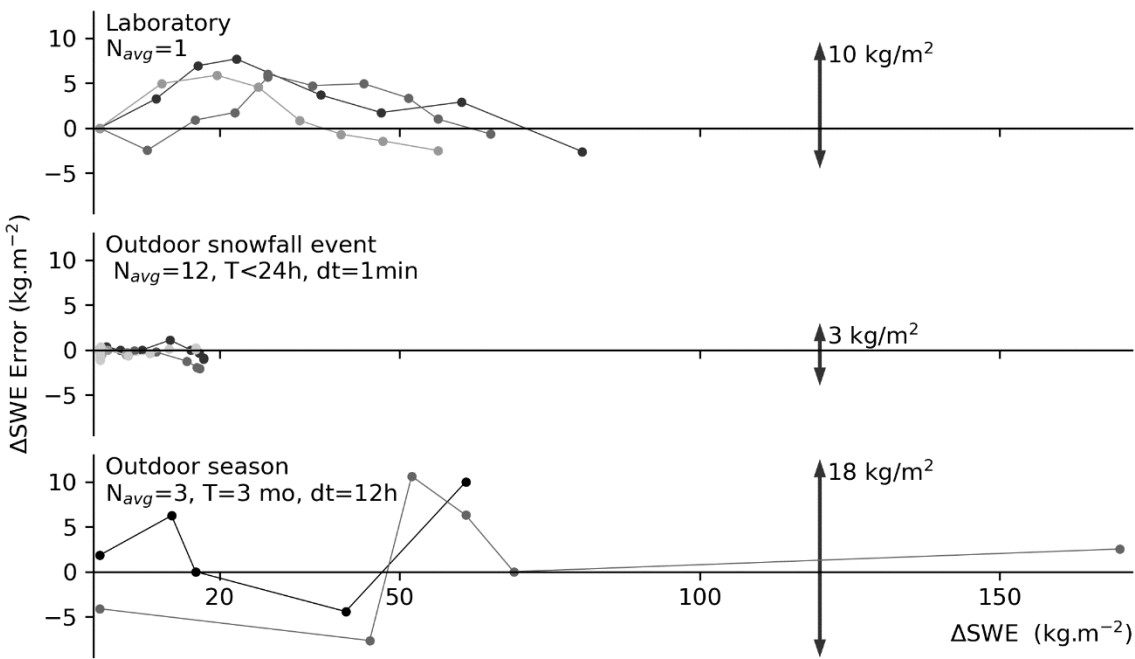

Fig. A1: Difference between the SWE measured by RFID and by weighing, in the laboratory, during snowfall events, and throughout the season (periods 2 and 3 are used, because snow pit weighing surveys were available). ΔSWE represents the variation in SWE measured with the same calibration. Darker curves represent earlier measurements.

## Appendix 2: Temperature measurements

The temperature data was first calibrated, by setting the temperature to 0 °C on tags covered by wet snow. In wet snow, these tags displayed tag a constant temperature near 0°C (indicating wet snow), preceded and followed by distinct patterns of temperature variations compared to the highest tags in the air. It occurred on 2019-12-14 and 2020-03-10 during more than 8h, for the eight tags up to 53 cm. A second calibration step was performed on the other tags, between 2019-11-11 and 2019-11-14 at 20:00−06:00 each day when the snow was low, by fitting their intercept of a linear regression with the tags previously calibrated at 0°C.

In terms of accuracy, the tag's microcircuit manufacturer indicates a maximum error of ±2 °C before calibration, and ±1.2 °C after offset calibration, for temperatures within the range −40 °C to +60 °C. In our hands, the error before calibration was ±0.8 °C within the range −7 °C to 0°C. Calibration reduced the uncertainty to ±0.25 °C (Fig. A2), which corresponds to the numerical resolution (see Fig. 10). No drift or random noise was visible.

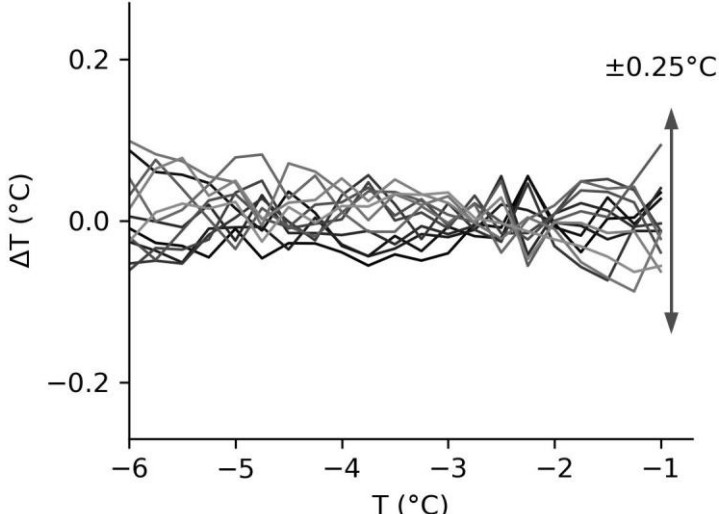

**Fig. A2: Difference in the temperature ΔT measured by the tags at a height of 83-163 cm, and their average measurement after calibrating the offset. The data was measured during the period of the second calibration step. It shows that there is no need for a 2-point calibration (=the measurement slope) on each individual tag.**

The tag temperature was plotted alongside the air temperature, and the snow surface temperature (Fig. A3 for each tag up to 0.64 m, then average for all tags >0.68 m (always above snow)). The temperature recorded by tags above the snow level correlated well with the air temperature. Tag temperature was higher than air temperature in the sunlight and lower at night due to radiative heat transfer, temporary snow/ice accumulation on tags, and to heat conduction through the tag support. For tags present in the snowpack, temperatures remained ≤ 0 °C, and no correlation with air temperature was observed. The temperature measurements confirmed that snow melted around the tag poles just before 2019-12-19 and 2020-02-03. Indeed, on 2019-12-21, the snow depth was indicated as <0.18 m based on the tag's temperature; measurement with a laser sensor indicated a depth of 0.25 m. On 2020-02-06, the snow depth determined based on tag temperature was <0.33 m; and 0.6 m according to the laser sensor. The snow depth offset thus appears to have accumulated after both accelerated melting events. As another indicator,

a stable temperature near 0 °C indicates that the snowpack is partially wet near the measuring tag (for example on 2020-03-10, up to 38 cm). During these periods, the temperature measured remained within 0 °C ±1 °C, which is consistent with the accuracy given by the manufacturer. Tags close to the ground remained around 0 °C most of the time, indicating that snow near the ground stays wet. Again, this behavior is expected to be due to heat transfer from the ground. However, the snow near the ground should remain only slightly wet most of the time because the heat flux coming from the ground is small compared to the heat needed to melt frozen water. After 2020-03-23, once the snowpack had entirely melted near the tags, the temperature of the lowest tags increased above 0 °C, as expected. These results confirm that RFID tags can monitor and spatialize temperatures, opening another perspective for the use of RFID tags to monitor the snowpack (e.g., Bagshaw et al., 2018).

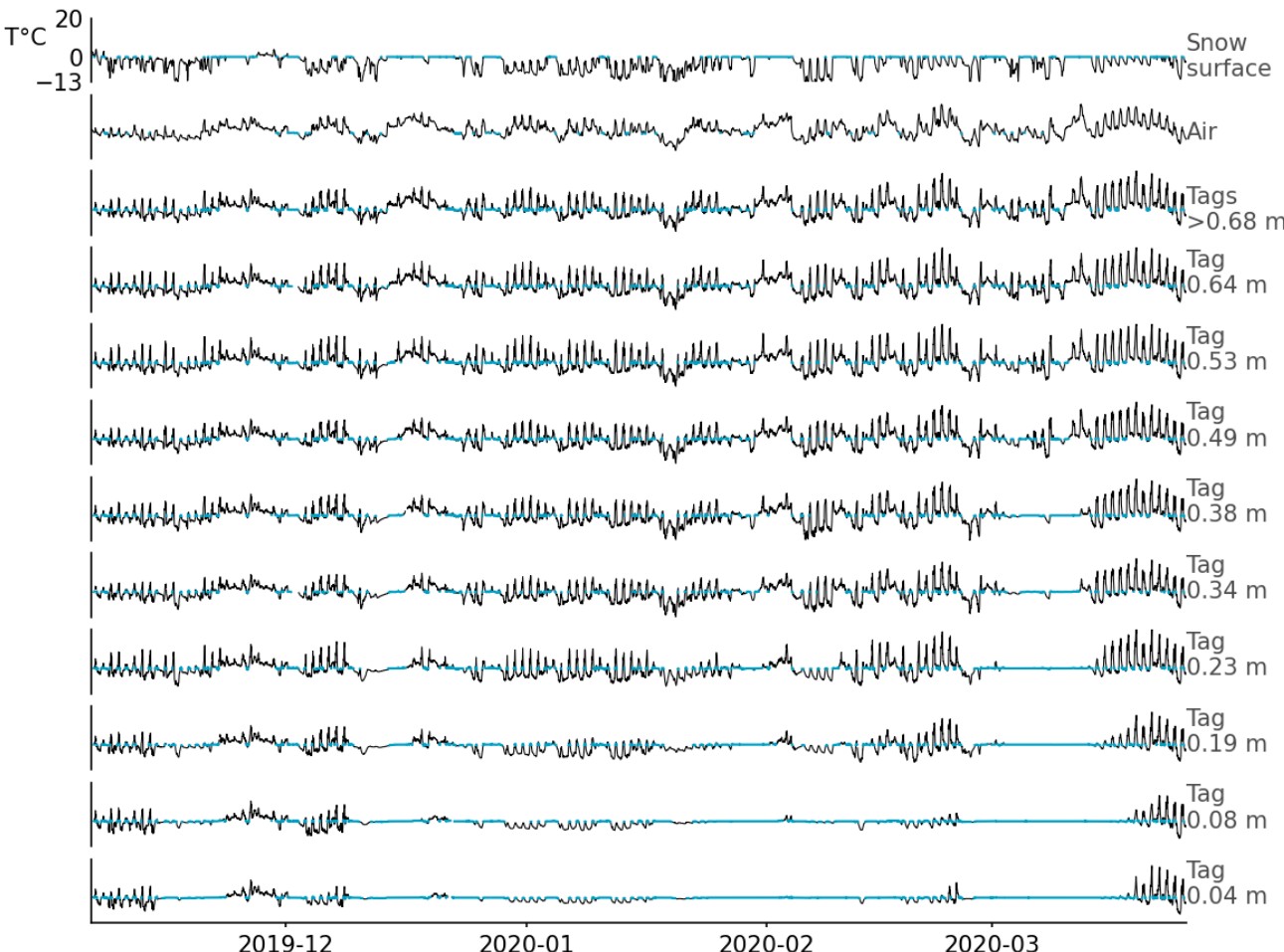

**Fig. A3: Temperature measured by RFID tags, from 0.04 to 0.64 m above ground, and the average of tags above 0.68 height. The air and snow surface temperatures were measured by independent instruments. The lines in blue represent T=0 °C ±0.3 °C.**

## Appendix 3: Detail on the tags

For this study we used Survivor B battery-powered tags because we were accustomed to these devices, and because of their long read-range. A picture of the tag, and the inside after removing its casing, is shown in Fig. A4. We want to emphasize that (1) the method presented works with any backscattering RFID tag, provided the signal's

phase can be read, and (2) the method works also works without battery, but only with a lower read-range. Readers who wish to reproduce the experiments could use any tag with a long read-range, whether batteryless or battery−assisted.

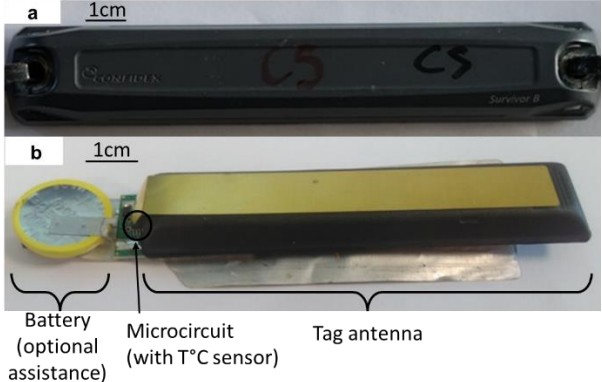

**Fig. A4: The commercial tag used in the study, (a) in its casing, and (b) without its casing. The battery is optional, but was used here**
**to maximize read-range performance. The method can be replicated with any batteryless tag for the SWE. It requires specific sensing tags (with or without battery) to monitor temperature, available from any RFID reseller.**

**Appendix 4: Interim results and wet snow periods**

We present interim results and detail some corrections required compute the SWE over the whole winter season (2019–2020) at the Col de Porte. The raw indicator of SWE variations is shown in Fig. A5 after unwrapping, but before removing wet snow periods, recalibrating due to melting, and averaging multiple tags. The SWE measurement based on cosmic rays data is also presented, with manually weighting of the snow pits (Lejeune et al., 2019). In addition, the snow depth (measured with a laser, in the pits, and from a visual pole), the lowest temperatures for each day (air, tags above snow, and snow surface), and the daily precipitation (with an estimation of the solid-to-liquid ratio) are indicated. The solid-to-liquid ratio of precipitation was obtained by estimating wether the precipitation should contain 0%, 50% or 100% liquid water, based on air temperature, snow radiations and expertise, for each hour of precipitation. The resulting quantities of liquid and solid water was cumulated every the day. The unwrapped indicator of SWE variations obtained from the three tags (Fig. A5.a, continuous lines in light colors) correlated visually with the reference SWE. As expected, the unwrapped phase returned to close to its initial value  at the end of the season.

The presence of liquid water in the snow also modifies the phase delay, and would not be differentiated from an increase of SWE. Liquid water affects the phase delay both by slowing the wave transmitted through the snowpack (e.g., Bradford et al., 2009; Tiuri et al., 1984) and by coupling with the tag antenna (Caccami et al., 2015; Le Breton et al., 2017; Dey et al., 2019). We identified dry snow periods from their constant or slowly evolving phase delay— occurring typically from 00:00 to 07:00. In contrast, the phase delay changed constantly with wet snow, due to its unstable snow liquid water content (wet snow either melts or refreezes).

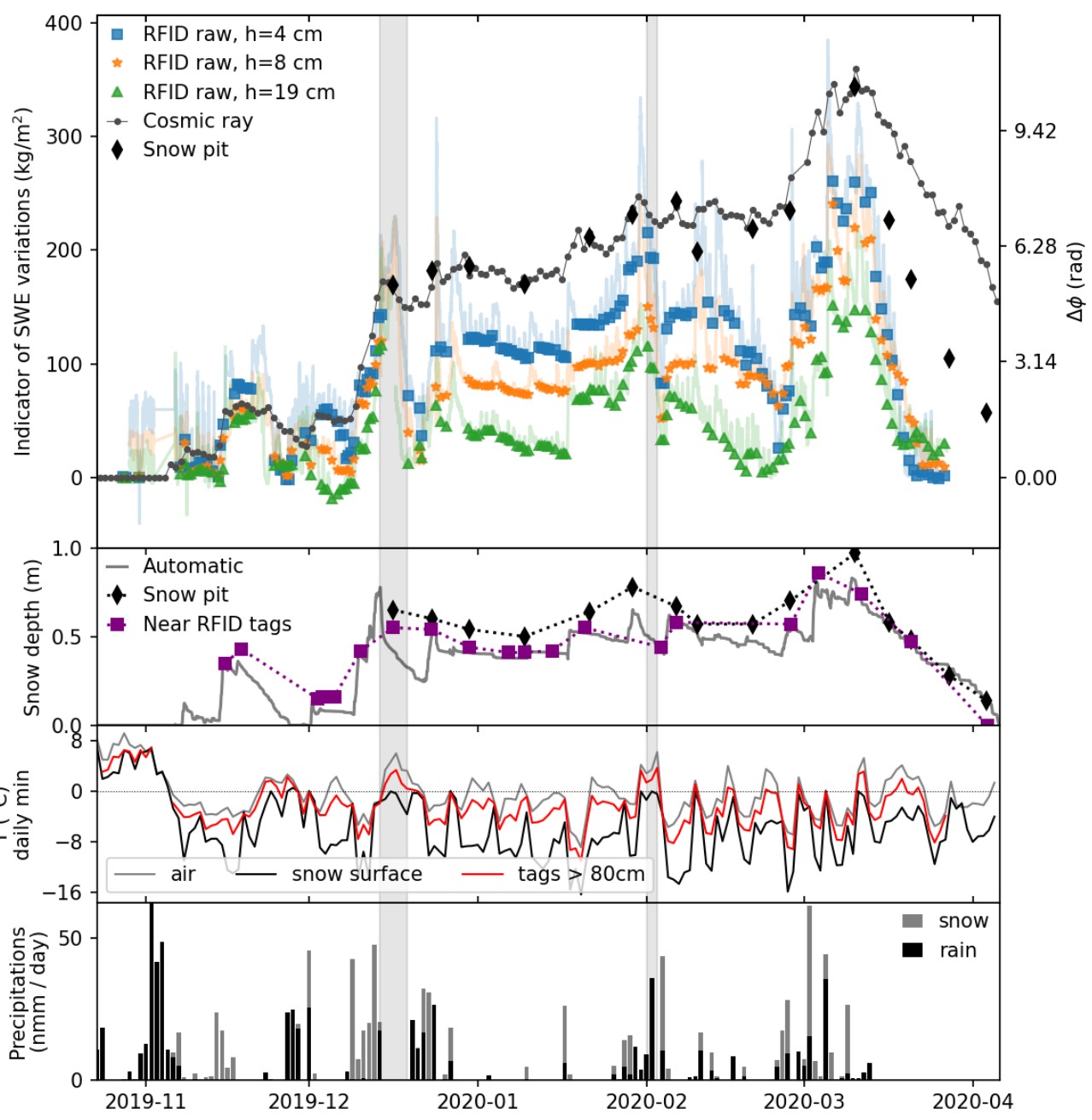

**Fig. A5: Raw indicator of SWE variations, with their equivalent variation of phase delay, for the snowpack located above the tags at 4, 8 and 19 cm from the ground. Periods of wet snowpack (peaks on the raw SWE indicators) were removed, and only the colored markers were considered when estimating the SWE. The SWE was also measured by automatic cosmic ray neutron counting and from snow pit surveys. The figure also shows the snow depth, daily minimum air temperature, and precipitation. In the grayed periods, a reheat accelerated snowpack melting around the tag support.**

### Appendix 5: Recalibration due to reheat

The step 5 in the Sect. 2.3 workflow was introduced to mitigate the acceleration of snowmelt caused by the installation. This effect occurred twice during the winter (from 2019-12-14 to 2019-12-19 and from 2020-02-01 to 2020-02-03), after strong wet precipitation combined with an air temperature that remained >0 °C over several days (Fig. A5), limiting the nightly refreezing. The influence was likely due to the thermal bridge and preferential melt-water path through the snow, caused by the tag support. The resulting increase in snowmelt was observed on

photographs (Fig. A6), on the non-reversible offset formed both between the RFID and the reference SWE (Fig. A5), and on the offset between the snow depth and the variations in tag temperature (Fig. A3). To mitigate this effect, we distinguished the three periods starting on (1) 2019-10-23 (2) 2019-12-19 (3) 2020-02-03. In periods 2 and 3, we recalibrated the SWE by adding an offset to fit the value of a reference manual pit survey, marked as ref in Fig. 6 (on 2019-12-30 for period 2 and 2020-02-06 for period 3).

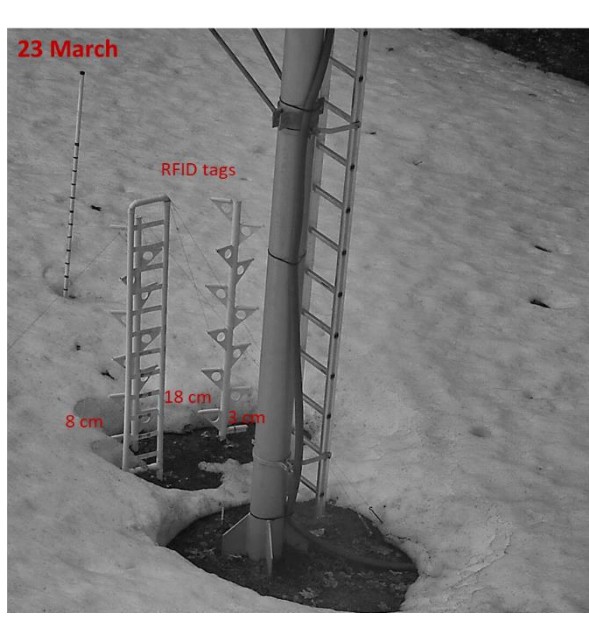

**Fig. A6: Photograph of the monitoring installation taken from the webcam, on 2020-03-23 at 12:00, confirming that the snowpack had melted faster around the tag supports, and that there was no more snow around the tags on this date.**

 **Appendix 6: Illustration of multipathing**

A simple experiment was done, in a similar configuration to the Col de Porte but at a different site, with dry snow. Instead of placing a vertical array of tags, the same tag was moved vertically in and above the snow (See Fig. A7c). The difference between the measured phase and the theoretical phase in free space (Fig. A7a), and the signal strength received (Fig A7b) revealed a clear oscillation. The period is half a wavelength ($\approx$17.4 cm in the air). Its influence on the phase and received signal strength reaches up to $\pm$2 rad and $\pm$10 dB (with one peak at $-$45 dB inside the snow). These results illustrate the effect of multipathing, and its spatial variability. A communication on this topic is in preparation.

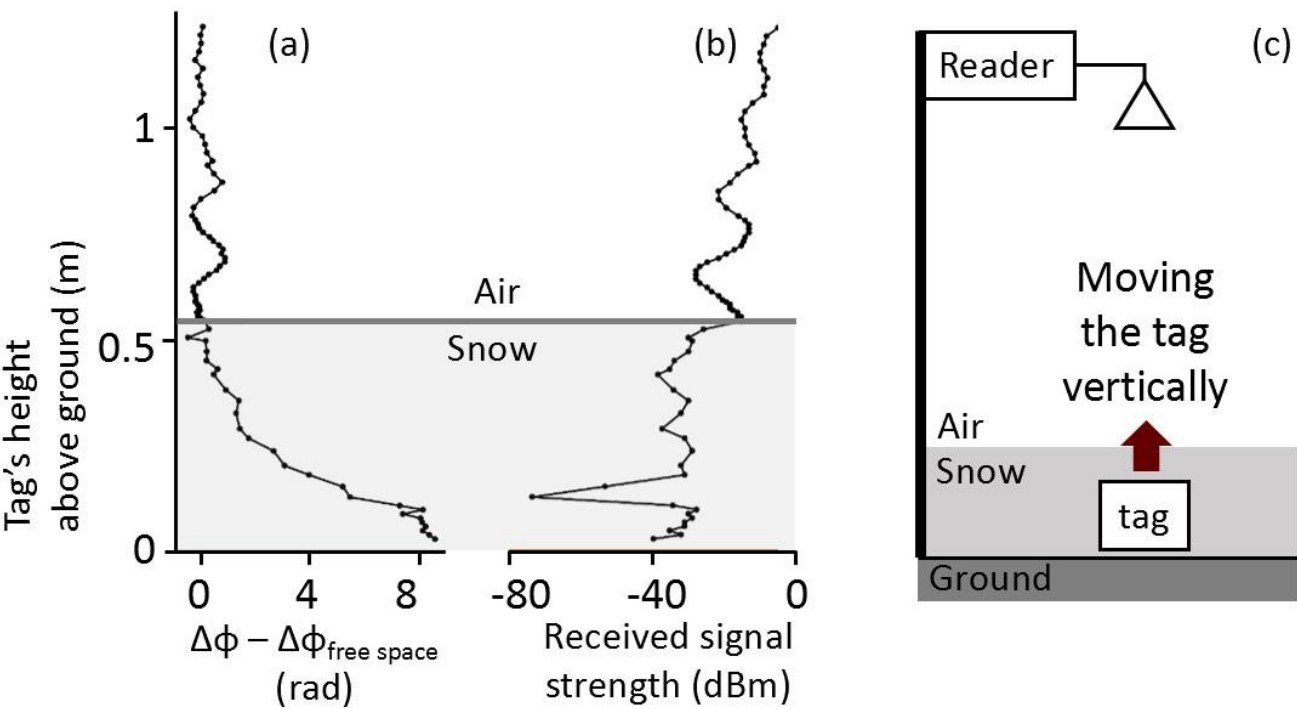

740

**Fig. A7: Simple experiment to illustrate multipathing. A tag was moved above and under dry snow, with the reader located above the snow. The results present (a) the difference between the theorical phase in free space and the measured phase, (b) the received signal strength.**