# Peer review of "Snow water equivalent can be monitored using RFID signal propagation"

_EGUsphere, 2022_

## Referee Comment (RC1)

egusphere-2022-761, "Monitoring snowpack SWE ....",        Review 2022-09-08

General comments:
1) In this manuscript, the authors propose and test a new method for SWE estimation: the use of passive (or semi active?) tags for RadioFrequency IDentification (RFID) in a narrow frequency band around 866 MHz. Due to problems and difficulties with other methods used so far, as the authors correctly describe in the introduction, the use of RFID could be highly attractive. The frequency range is optimum with respect to penetration through any seasonal snowpack, primarily for dry snow. Even for wet snow the method may be used if the liquid-water content is known (and not too large).
2) The main difficulties are related to the determination of the total phase due to the short wavelength in comparison with the length of the propagation path, resulting in phase ambiguities. This problem can be solved by phase unwrapping if continuous observations exist. The authors found and described ways to overcome problems with short data gaps.
3) An unexpected problem came up by large phase uncertainties due to interference by multipath propagation of the sensing waves. The use of independent measurements helped in reducing these uncertainties. In my opinion, the problem is still severe and should be improved. The questions:
 - what causes multipath effects?
 - how can they be reduced?
have not been addressed by the authors. Ways to tackle these questions are by numerical simulations using a forward model or by experimental work.
4) No information is given on the properties of the antennas used.
5) No information is given on the scattering and absorption cross sections of the tags used, nor of the supporting structures.
6) No information is given on the method used to discriminate the responses and the backscattered signals from different tags, and how this discrimination may be linked with the phase determination.
7) Information is also missing on how the temperature measurement of the tags is working.
8) An alternative to the vertical stack of tags would be tags close to the ground surface in a type of (phased?) array. The tags close to the ground are the ones that gave most of the information.

Details:
1) English language should be improved.
2) Improve the final part of the Introduction, sentence on lines 74-75, and "Section 0", lines 85 to 88.
3) After Equation (1): "in-phase and quadrature" are generally used to describe the complex electrical field of electromagnetic waves. Don't use it here for the complex dielectric constant.
4) Section 2.1 **Theory: from phase delay to SWE** to be improved and simplified as a whole.
5) Section 2.2 **Instrumentation** does not present the instruments and their properties. I missed this description, see general comments above. The subsection describes experiments and sites.

---

## Author Comment (AC3)

Answer to referee's comments, on the manuscript «Monitoring snowpack SWE and temperature using RFID tags as wireless sensors»; 2023-01-09; Referee 1.

**General comments:**

**1.1 In this manuscript, the authors propose and test a new method for SWE estimation: the use of passive (or semi active?) tags**

There are still discussion on the denomination because "passive" and "battery" is usually opposed. The right denomination to us is "passive battery assisted" tag, because from the outside the tag works the same as passive tags, only internally it has a boost of energy. Active tags are totally different: they do not use the widely used standards of passive RFID, they generate their own wave, they consume much more energy, etc…. Our method works both with passive batteryless or passive battery-assisted tags (only with different performance in the read range), but would not work with active tags.

We have added a clarification on line 135–140.

**1.2 An unexpected problem came up by large phase uncertainties due to interference by multipath propagation of the sensing waves. The use of independent measurements helped in reducing these uncertainties. In my opinion, the problem is still severe and should be improved. The questions:**

- **what causes multipath effects?**
- **how can they be reduced?**

**have not been addressed by the authors. Ways to tackle these questions are by numerical simulations using a forward model or by experimental work.**

We agree that there is a need to better understand the multipathing effect on RFID systems in snow contexts. However, this is far beyond the scope of the present study (multipathing is a general issue in RFID, particularly for localization indoors where multipathing can be strong). We have a work in progress on this topic, for which we will dedicate a communication in itself. Two results are presented below in this document.

To reduce the multipathing in the present study, we then suggest to use an array of tags placed very close to the ground. It should help average the spread due to spatial diversity, and should also reduce the reflection between tags, and the reflection on the supporting material. Given that anyway we want to remove the vertical support to reduce the influence of the system on the snowpack.

What causes multipathing:

- The reflection on:
  - The tags
  - The snow surface
  - The soil surface
  - The different layers of snow
  - The light plastic supporting structure that holds the tags
  - The large metallic supporting structure that holds the reader

What causes variations of multipathing over time:

- Changes either in amplitude or in phase delay of the reflected paths, due to variations in height, density and moisture content of the snow.

In the following experiment, we have moved a tag at different heights above and below the snowpack surface. We represent the difference between the measured phase and the phase expected if the tag moved in free space in the air.

We can see clearly the effect of multipathing when the tag is above the snowpack, with variations of 1.4 rad amplitude, that correspond to the spread observed in the present study (up to 1.2 rad).

[Figure]

In another experiment, we have removed a thin layer of snow from the ground, with tags placed at different locations. As a consequence, the phase varied between +1 to -2 rad (3 rad of amplitude). It is coherent with the changes expected from a simple 3-ray model.

[Figure]

**1.3 No information is given on the properties of the antennas used.**

We have added the exact model number which allows to find the data sheet. We corrected the gain, and added the beam width, polarization, and protection indice (Line 130–135).

**1.4 No information is given on the scattering and absorption cross sections of the tags used, nor of the supporting structures.**

We are not sure of what it would bring in the experiment's context. I could see two goals, either for tag designing, or for characterization of the multipathing.

Concerning the tag design and behavior: the principle of this experiment is independent of the tag used (at the contrary to the RSSI for example, but we did not want to display or discuss RSSI data in this article, in order to stay focused). We can display the tag's response depending on the frequency (see figure below for example), but again, this study is not about the tag's design.

[Figure]

Concerning the multipathing: as we suggest in 1.2, multipathing is one of the major issue in RFID in general, particularly for the localization methods from which our approach originates. If we wanted to model the multipathing, we would need the cross-sections. But then, it relates to comment 1.1, and understanding the multipathing is also out of this study's scope.

We have added a picture of the tag without its casing.

**1.5 No information is given on the method used to discriminate the responses and the backscattered signals from different tags, and how this discrimination may be linked with the phase determination.**

Indeed, it needs explanation. The tag being interrogated by the reader modulates its reflection between two states, while the other reflections (environment and tags) are static. The phase difference of arrival is calculated from the IQ difference between the two states, which relates only to the interrogated tag (given a static environment).

I have explained on lines 140–145, and added a reference to Nikitin 2010 for further understanding.

**1.6 Information is also missing on how the temperature measurement of the tags is working.**

The temperature sensor in included in the tag's microcircuit, that we use as it is.

We have added a picture of the tag, in order to see the location of the microcircuit that measure its temperature.

We have also added a reference to the integrated microcircuit model datasheet.

**1.7 An alternative to the vertical stack of tags would be tags close to the ground surface in a type of (phased?) array. The tags close to the ground are the ones that gave most of the information.**

Absolutely! We are planning this update in our next installation.

→I have added this suggestion more clearly on line 257 and 267.

**Details:**

**1.8 English language should be improved.**

We are working on it.

**1.9 Improve the final part of the Introduction, sentence on lines 74–75, and "Section 0", lines 85 to 88.**

corrected

**1.10 After Equation (1): "in-phase and quadrature" are generally used to describe the complex electrical field of electromagnetic waves. Don't use it here for the complex dielectric constant.**

Ok, modified.

**1.11 Section 2.1 Theory: from phase delay to SWE to be improved and simplified as a whole.**

We are simplifying it.

**1.12 Section 2.2 Instrumentation does not present the instruments and their properties. I missed this description, see general comments above. The subsection describes experiments and sites.**

We have increased the description of the instruments.

---

## Author Response (AR1)

**Message to the reviewer and editors**

Dear editor, and dear reviewers. First, thank you very much for your comments and suggestions that guided me to reading more, improving details, and, I hope, greatly improving this manuscript. Beyond the answer of your comments (all addressed), I also wanted to convey a clearer message in the manuscript, and made it much more concise. The changes were so important that highlighting them made does not make sense. Let me instead list you the modifications here:

- I focus on the SWE measurement. The temperature becomes secondary. The title changed.
- I placed all the results that distract from the main message (but are necessary for the rigor) to the appendix. Fig A3, Fig A5, Fig A6 were in the main text of the previous version.
- I have added the discussion part, and table 1 which describe the performance of our method using Royer 2021's criteria.
- All the figures were redrawn to be more readable.
- All the previous text was more concise
- New section 3.4, specifically on accuracy estimation.
- I have added appendix 1 about SWE uncertainty calculation
- I have moved the temperature results to appendix 2, and added a part on calibration and uncertainty estimation.
- I have added the appendix 3 to describe the specific tag we used.
- The intermediary results of the season were moved in appendix 4, and the description of the technical issue of accelerated snowmelt was moved to appendix 5.
- I have added the appendix 6, with a simple experiment on multipathing.
- I have removed old table 1 which compare all instruments in the intro. The key gaps to fill are in the intro. And another table 1 dealing only with our method is in the discussion.
- I have removed table 2 providing details on snowfall events, too cluttered. Instead I inserted the same uncertainty information (but not only) on Fig A1 in appendix 1. This data was also improved, see below.

There were changes on the data as well:

- I now use the mean value between tag everywhere (no more median)
- I reduced a bit the overestimation of the density due to snow settling by extrapolating the rate of snow depth decrease caused by settling after precipitation (Fig 7 and Sect. 3.4)
- For the same reason, when two snowfall occurred consecutively, I kept only the first one. Thanks to this, I could include all the four snowfall events in the Fig. 7, instead of keeping only 2 events for density estimation in the previous manuscript.
- By rechecking the snowfall events (Fig. 5) I saw, from their temperature, that some tags were no more in the snow despite a higher indicated snow depth (due to SWE offset caused by melting). So I used the 4 or 5 lowest tags, instead of 6 previously. (results are better, but there are still big outliers, that we attribute to multipathing: this conclusion does not change.)
- After remeasuring dimensions from photos, the height of the lowest tag appeared closer to 4 cm than to 3 cm indicated in previous manuscript, so I added +1cm on all the tags from this stick. It doesn't change any result anywhere.
- I have completed fig. 7 on the density of all the data: I have included density estimation for each hour of the snowfall events in which precipitations occurred and displayed them separately. I have compared the weighting and RFID measurements of each pit survey.

You can find below the answer specific to your comments.

Thank you again for your revisions

Mathieu Le Breton

**1  Answer to the editor**

Dear Dr. Mathieu Le Breton,

we have received two valuable reviews for this manuscript. While the reviewers find the work in general interesting, they raise important issues, especially to explain several points better. The reviewers recommend major revision, which goes in line with my own reading of the manuscript. The feedback provided in the reviewer assessments is important and should be taken carefully into account. In addition to the good points made by the reviewers, I have for now some points to be addressed carefully as well (see comments below).

As a next step, I would like to invite you to revise the manuscript.

Best regards,

Franziska Koch

General comments:

—Please use for snow depth units metres or centimetres (not mm as you did for example in Figure 4 or Figure 5).

**I have changed Figures 4 and 5, and a few units in the text.**

—Any comments on the maximum SWE to be monitored with your method?

**A discussion was added in section 6, line 370.**

**We estimate a maximum theoretical value of 3 000 kg/m2.**

—Regarding language issues, a thorough copy-editing by a (professional) native speaker is recommended.

**A professional proofreader revised the manuscript (we made few modifications after her corrections, but they were minor)**

—In the current version of your RFID measurement technique, you need some corrections to depict/process the "right" SWE (Section 3.3). Can you please discuss/outline in more detail, which could be a way to avoid such corrections (e.g. fixing to values of manual snow pits [l. 249], picking carefully the wet/dry periods…) in future to have a fully independent technique?

**I had initially added a specific discussion on this, but it appeared vague, because I have intuition on how to automatize, but nothing is sure before really doing it. To provide more details, I have added the section 2.3 describing the processing workflow that we have used (line 152).**

—Section 3.3. (especially to get to your final result in Figure 8) is difficult to follow and understand. Please revise in terms of explaining all steps for a better understanding.

**The section 3.3 was removed, and transformed in a more concise processing workflow (section 2.3)**

Specific comments:

—l. 27: SWE "expressed as surface density" is misleading. Please reformulate.

**Done.**

—l. 40ff and further locations in the manuscript: Regarding GPS/GNSS please be consistent, use either or—I would recommend to use GNSS.

**Done. I use only GNSS.**

—l.45: It is true for upGPR setups, that they are expensive; however, this is not true for low-cost GNSS setups, especially when you just consider the low-cost equipment—which would be in the range or even in a lower than the RFID tags and antenna setup (power supply, management unit, processing of data etc. not included…). So please just mention upGPR in the context of being expensive.

**Done. I have also completed the comparison with GNSS, GMON and CNRP in discussion, leading to improving the introduction too.**

—l. 46: It should be upward-looking GPRs

**I removed the distinction between upward and downward-looking GPRs, because I do not compare with GPR in the discussion. (I compared only with GNSS, GMON and CNRP compared by Royer 2021).**

—Table 1: Regarding the row GNSS please change the following in the column "area": m² (larger areas would correspond to snow depth recordings via GNSS reflectometry).

**I have removed this table to be more concise. A table 1 dealing only with our method, and a discussion, deal more precisely with the different criteria.**

**I also removed the mention to reflectometry in intro, as measuring snow depth is not our message.**

—l. 103: The permittivity range corresponds to dry snow. Please add "dry". It can be certainly higher for wet snow.

**Done. Line 75.**

—l. 141f: What is the size of the RFID tags?

**Size mentioned at l. 94**

Please mention also the temporal resolution of your final SWE measurements (1 min? 6 hours? daily?—was not so clear to me in the text).

**Added in the processing workflow, line 191.**

**Also discussed in the discussion, because the high time resolution (only with dry snow at this stage) is one of the main advantages of the method.**

—l. 145: Which depth did the snow layers have in your experiment? This is also unclear in l. 175f.

**I have renamed the x-axis as "snow depth" on Fig. 4, and use the same term on line 118.**

—l. 150f: Is there a reason why you chose these densities? Why did you not choose for example fresh snow with approx. 100 kg/m³ in your experiment?

**It was the best range we could obtain in the laboratory and with the snow we had in stock (=not fresh snow), without compressing the snow. The density was increased by changing the sieving size and by reusing the same snow across experiments.**

**Described on lines 117–118.**

—l. 154: Please give some more information on the two vertical arrays (diameter of poles, length of arms where RFID tags were installed on etc.).

**I have added information on the setup dimension, lines 130–134.**

**However, after discussing with C. Matzler, it appeared he wanted specific info on the geometry for modeling the entire multipathing setup. We added a discussion on multipathing (explaining with references that modeling the entire multipath channel seems too complex).**

**Given the issues of melting and multipathing, we clearly think that our geometry should be improved in the future, and not reproduced as it is (which we state in the discussion).**

Please discuss at a later point in the manuscript also more in depth potential uncertainties regarding the installation on the vertical arrays.

**I am not sure to have fully understood this comment. Here are some potential elements:**

- **line 134: added the description of rigging strings to avoid movements of the array.**
- **Line 321: its potential role on multipathing.**
- **Appendix 5, line 711: issues of recalibration (but nothing new here)**
- **line 330: one conclusion is that using an array of tags (whatever its geometry) improves accuracy.**
- **in conclusions, line 416: future developments should aim to improve tag array design.**

—l. 167: I guess you rather mean the monitoring of the melt out of individual RFID tags? Please clarify.

**Exactly. It is clarified. Lines 711 and 719.**

—l. 200ff: Please explain better how the final (median) SWE value was derived. Not sure, if I understood correctly, but did you calculate the median of the tag readings at different installed height.

**Clarified in Section 2.3, on lines 156–160, and 186–189.**

**I now use only the mean value (it does not fundamentally change the result, but it's more rigorous and simpler to use the same method everywhere).**

If so, is it than valid to use a median calculation? Please clarify.

**I do not use the median anymore. Only the mean.**

—Figure 5: Why are the readings of the RFID tag at 13 and 28 cm installation height missing?

**There was no tag at 13 cm and 28 cm height. I looked if there was an error in the text, but did not find it. So I could not correct for this comment. Maybe the issue disappeared on this ms??**

… The connection with the colors regarding antenna 1 and antenna 2 is not easy to grasp. I suggest to use colors, e.g., in the shades of blue to green for connected to reader antenna 1 and e.g. yellow to red for reader antenna 2.

**I have greatly simplified these figures which were quite complex to grasp, by merging them (Fig. 5). I have also changed the colormap to distinguish the two antennas. It is still not perfect, but that is the best I managed to do given the numerous curves.**

—Figure 6: Why don't you include RFID readings from installation heights up to 43 cm in this figure as according to the measured snow height during the season these tags should also be covered by snow most of the time.

**We did not include the phase/SWE measurements of these tags (on the old Fig. 6, now Fig. A5) for several reasons.**

**1) Because in practice they were not covered by snow most of the time, due to accelerated snow melting around the tags pole (that is the reason for the two recalibrations during the winter). We can see this from the temperature measurements.**

**2) Furthermore, using lowest tags reduces the bias that could appear due to settling.**

**3) We did not use these tags to compute SWE over the entire season.**

Regarding the precipitation panel: How did you separate into rain and snow?

**I added description it on lines 681–683.**

—L. 280ff: Please relate the reported errors with the entire snow depth/SWE of the experiment or outdoor test.

**Comparison with snow weighting added on the new Fig. A1, and the new section 3.4.**

—L. 288f: Why did you expect that the temperature within the snowpack correlates with the air temperature?

**I meant the contrary, which was unclear due to poor English. Corrected.**

—Figure 10: According to Fig. 8, the snow depth is over a long period higher than approx. 33 or even 43 cm. However, looking at Figure 10, the RFID tags above an installation height of 18 cm seem not to be within the snowpack as they react similar to the air temperature. Please clarify.

**Indeed. That is because the issue of faster melting due to the vertical pole, that made us recalibrate the measurement twice in the season. I have clarified it on line 360:**

**That is stated in the appendices 4 and 5, and on the line 182.**

The 0 °C line in these plots is too weak and should be represented better in this figure. Moreover, please add units to the installation heights in the graph.

**Done. To highlight when tags are on the zero line, I displayed them in blue (Fig. A3).**

Revisions on manuscript: Snow water equivalent can be monitored using RFID signal propagation

**2   Answer to Referee 1**

General comments:

1.1 In this manuscript, the authors propose and test a new method for SWE estimation: the use of passive (or semi active?) tags

**There are still discussion on the denomination because "passive" and "battery" is usually opposed. The right denomination to us is "passive battery assisted" tag, because from the outside the tag works the same as passive tags, only internally it has a boost of energy. Active tags are totally different: they do not use the widely used standards of passive RFID, they generate their own wave, they consume much more energy, etc. Our method works both with passive batteryless or passive battery-assisted tags (only with different performance in the read range), but would not work with active tags.**

**Clarified on line 95–97, and on appendix 3.**

1.2 An unexpected problem came up by large phase uncertainties due to interference by multipath propagation of the sensing waves. The use of independent measurements helped in reducing these uncertainties. In my opinion, the problem is still severe and should be improved. The questions:

- what causes multipath effects?
- how can they be reduced?

**We have added a discussion on multipathing, line $314 - 332$.**

have not been addressed by the authors. Ways to tackle these questions are by numerical simulations using a forward model or by experimental work.

**We have added brief results from of another experiment that shows very clearly the role of multipathing, on appendix 6 and Fig. A7.**

**Discussion on multipathing is about this (lines 314–332). We disagree with this comment. We think, on the contrary, that a tag array is an efficient way to mitigate multipathing (and the ability to use an array is an advantage of the RFID method). In addition modeling the environment to make corrections is very complex, I am not sure it is feasible. At least, we know no study that has done it, despite multipathing being the issue n° 1 for indoors localization.**

1.3 No information is given on the properties of the antennas used.

**We have added information on line $92 - 95$.**

1.4 No information is given on the scattering and absorption cross sections of the tags used, nor of the supporting structures.

**This would be helpful to model the multipathing, which relates to previous comment. However, we think that is not feasible, at least not in our study. It is explained in discussion about multipathing in section 6.**

1.5 No information is given on the method used to discriminate the responses and the backscattered signals from different tags, and how this discrimination may be linked with the phase determination.

**I have explained it on line 107, with a reference for detailed information.**

1.6 Information is also missing on how the temperature measurement of the tags is working.

**Added: line 101: the tag embeds an integrated.**

**Fig. A4: Picture of the tag, with the location of the temperature sensor.**

1.7 An alternative to the vertical stack of tags would be tags close to the ground surface in a type of (phased?) array. The tags close to the ground are the ones that gave most of the information.

**Absolutely! We are thinking about this update in our next installation. However, the type of array geometry that would work better is still speculative before doing the experiment, so we did not suggest any particular geometry in the manuscript.**

**Added lines 357–358: "Absolute localization methods based on tag arrays (Xu et al., 2023) could also be investigated."**

**Added on line 417, the perspective: "Future developments should aim to improve tag array design."**

Details:

1.8 English language should be improved.

**The manuscript was made much more concise.**

**It was then proofread by a professional proofreader.**

1.9 Improve the final part of the Introduction, sentence on lines 74–75, and "Section 0", lines 85 to 88.

**Corrected.**

**I have revised the introduction deeply to make it shorter, simpler and to the point.**

1.10 After Equation (1): "in-phase and quadrature" are generally used to describe the complex electrical field of electromagnetic waves. Don't use it here for the complex dielectric constant.

**Ok, I have changed it.**

1.11 Section 2.1 Theory: from phase delay to SWE to be improved and simplified as a whole.

**I have simplified it. It went from 9 equations to 6 equations, and the text is more concise.**

1.12 Section 2.2 Instrumentation does not present the instruments and their properties. I missed this description, see general comments above. The subsection describes experiments and sites.

**We have completed the description of the instruments and their properties.**

**3   Answer to Referee 2**

This work proposes the use of RFID for monitoring the snowpack SWE and temperature. The topic and the application is interesting and the authors provided experimental results which seem to provide good results. However there are some criticism that should be clarified.

1) the authors claim to use standard RFID, that this is not really true a standard RFID do not be equipped with sensors, you must design a customized tag. The same for the battery you can't connect a battery to a standard tag. You should provide more information related to the Considered RFID schema.

**The SWE measurement of our method does use standard tags. That is the strength of this (new) propagation-based sensing method. It is derived from localization methods that work with any tag. You are right, however, that temperature measurements are only a few tag models (nevertheless bought on the shelf). Therefore**

- **I have put the focus on SWE measurement which is the strongest value and works with any tag based on ETSI 3028-208 standard. I have removed the focus from temperature (in the title and all the paper), which indeed requires a specific tag.**
- **I have added details on appendix 3 about the tag used, that was bought off-the-shelf.**
- **Line 395–398: I added a specific discussion section on this topic.**

2) you perform a measure of phase difference but it is not clear how. If you used inductive coupling rfid it is quite difficult. I suppose that you used a RF tag, and you claim that the reader is able to detect the phase difference. Can you please better explain how? You have to analyse the signals at the demodulator in order to detect the phase difference of did you use another technique? Please explain.

**We used an UHF RFID (working around 868 MHz). The reader (Impinj R700) has the capacity to read phase difference of arrival when reading a tag, out-of-the-box. Several readers have this capacity, such as the ImpinjSR420, Impinj R700, or the ThingMagic M6.**

➔ **I have added a reference those studies on the phase measurement techniques:**

1. **Miesen, R., Parr, A., Schleu, J. & Vossiek, M. 360° carrier phase measurement for UHF RFID local positioning. in 2013 IEEE International Conference on RFID-Technologies and Applications (RFID-TA) 1–6 (2013). doi:10.1109/RFID-TA.2013.6694499.**
2. **1. Nikitin, P. V. et al. Phase based spatial identification of UHF RFID tags. in IEEE Int. Conf. RFID 102–109 (IEEE, 2010). doi:10.1109/RFID.2010.5467253.**

3) did you take into account the attenuation introduced in the RF signal by the snow?

**This study exploits only the delay. Yet I mentioned the attenuation:**

**—Lines 343–345: I suggest that we could use attenuation for measurements in the future.**

**—Lines 370–386: I added a discussion on the sources of attenuation, that are related to the tag's read range, in order to estimate the maximum snow depth.**

4) I suppose that you modified the reader, don't you? If yes please report the introduced customisations.

**No, we have used a commercial off-the-shelf reader, not customized. The method works with any reader that is able to read the phase difference of arrival.**